# Clustering uncertain overlapping symptoms of multiple diseases in clinical diagnosis

Asif Ali Wagan, Shahnawaz Talpur and Sanam Narejo

Computer Systems Engineering, Mehran University of Engineering & Technology Jamshoro, Jamshoro, Sindh, Pakistan



## ABSTRACT

In various fields, including medical science, datasets characterized by uncertainty are generated. Conventional clustering algorithms, designed for deterministic data, often prove inadequate when applied to uncertain data, posing significant challenges. Recent advancements have introduced clustering algorithms based on a possible world model, specifically designed to handle uncertainty, showing promising outcomes. However, these algorithms face two primary issues. First, they treat all possible worlds equally, neglecting the relative importance of each world. Second, they employ time-consuming and inefficient post-processing techniques for world selection. This research aims to create clusters of observed symptoms in patients, enabling the exploration of intricate relationships between symptoms. However, the symptoms dataset presents unique challenges, as it entails uncertainty and exhibits overlapping symptoms across multiple diseases, rendering the formation of mutually exclusive clusters impractical. Conventional similarity measures, assuming mutually exclusive clusters, fail to address these challenges effectively. Furthermore, the categorical nature of the symptoms dataset further complicates the analysis, as most similarity measures are optimized for numerical datasets. To overcome these scientific obstacles, this research proposes an innovative clustering algorithm that considers the precise weight of each symptom in every disease, facilitating the generation of overlapping clusters that accurately depict the associations between symptoms in the context of various diseases.

Corresponding author
Asif Ali Wagan,
asif.wagan@admin.muet.edu.pk

## INTRODUCTION

Advancements in technology have led to a significant increase in data generation across various fields, including medical science, wireless sensor networks (WSN), data integration, and information extraction (*Sharma & Seal, 2021a*). Clustering techniques have found widespread application in diverse domains such as economics, healthcare, and science, enabling the extraction of actionable knowledge (*Saxena et al., 2017*). However, the effectiveness of conventional clustering algorithms is hindered when faced with uncertain data, which is characterized by its inherently random nature (*Liu et al., 2021*).

While different modeling and querying techniques have been developed to handle uncertain data, clustering techniques have yet to reach their full potential in effectively addressing uncertainty (*Liu et al., 2019*). Among the clustering algorithms proposed for uncertain data, those based on the possible world model have shown promising outcomes. However, two key challenges persist with these algorithms. Firstly, they assign equal weight to all possible worlds, including the marginal worlds, resulting in the equal treatment of less important records. Secondly, the selection of true worlds necessitates the use of post-heuristic techniques, which can be time-consuming and inefficient (*Liu et al., 2019*; *Sharma & Seal, 2021a*).

In addition, clustering techniques have proven valuable in uncovering structural patterns within medical datasets. Notably, researchers have focused on clustering symptoms, as demonstrated by *Baden et al. (2020)* who investigated the clustering of symptoms such as pain, fatigue, and depression in survivors of prostate cancer. Similarly, other studies have examined symptom clustering in the context of depression (*Maglanoc et al., 2019*). Clustering symptoms into specific groups reveals strong relationships between them, offering valuable insights into diseases and assisting in the development of treatment strategies (*Yifan et al., 2020*). However, many medical datasets inherently contain overlapping information (*Khanmohammadi, Adibeig & Shanehbandy, 2017*). For instance, certain symptoms, such as poor muscle tone, may be shared among different diseases like hypothyroidism and Pompe disease. Additionally, patients may experience multiple coexisting conditions, such as diabetes and hypertension, further complicating the clustering process (*Brancati et al., 2019*). Consequently, there has been a growing interest in developing overlapping clustering techniques. Nonetheless, the presence of overlapping symptoms across multiple diseases introduces uncertainty. Moreover, traditional similarity measures, such as geometric distances used in clustering algorithms, struggle to effectively capture and cluster overlapping information (*Sharma & Seal, 2021a*). Geometric distance-based similarity measures are unable to fully capture the relationship between uncertain data and their distribution when extensive overlaps occur (*Sharma & Seal, 2021b*). Furthermore, the primary objective of similarity measures is to group points with the highest similarity into mutually exclusive clusters. However, when symptoms overlap across multiple diseases, they cannot be unequivocally assigned to a specific disease cluster. As a result, the existing clustering algorithms for uncertain data fail to generate overlapping clusters (*Gates et al., 2019*).

Categorical symptom datasets pose unique challenges in clustering analysis. Effective clustering techniques heavily rely on suitable proximity measures for computing (dis) similarity between data objects. However, applying similarity measures designed for numerical data to categorical datasets often results in suboptimal outcomes. Studies have underscored the limitations of traditional measures and emphasized the importance of tailored similarity metrics to address the specific challenges of clustering categorical data. It is evident that traditional similarity measures, predominantly designed for numerical datasets, may inadequately capture the underlying relationships and similarities within categorical symptom data.

This research addresses the challenges of clustering uncertain data, particularly in the context of medical datasets. Traditional clustering algorithms designed for certain data struggle to handle uncertainty effectively. Existing techniques for uncertain data show limited efficiency in producing optimal results. Overlapping symptoms in medical datasets further complicate the clustering process, as traditional similarity measures fail to capture and cluster this overlapping information accurately. This research aims to advance the understanding of clustering uncertain data and improve the accuracy of clustering results in medical datasets.

## PRELIMINARIES

### Possible world model

Machine learning algorithms like PwAdaBoost handle uncertain data using the possible world model (*Liu, Zhang & Zhang, 2019*), employed in various algorithms such as RPC and outlier-robust multi-view clustering (*Liu et al., 2021*; *Sharma & Seal, 2021a*). These algorithms assign weights to probability values to manage uncertain data complexities. Further research is needed to enhance this modeling approach.

To understand the possible world model, consider UD as an uncertain dataset with m attributes. Each attribute, denoted as $A_{ij}$, represents an uncertain numerical or categorical value for the $j$-th object. Uncertain numerical attributes are modeled using a probability density function $f(x)$ in the interval $[A_{ij}^l, A_{ij}^r]$, satisfying conditions (*Liu, Zhang & Zhang, 2019*):

$$f(x) \geq 0 \quad \forall x \in [A_{ij}^l, A_{ij}^r],$$

$$\int_{A_{ij}^l}^{A_{ij}^r} f(x), dx = 1. \tag{1}$$

For uncertain categorical datasets, $A_{ij}$ is modeled as a random variable in the discrete domain using a probability mass function $p$, with possible values $dom = \{v_1, v_2, \ldots, v_n\}$, satisfying (*Liu, Zhang & Zhang, 2019*):

$$P(A_{ij} = v_k) = p_k, \quad 1 \leq k \leq n,$$

$$\sum_{k=1}^{n} P_k = 1 \quad \text{for } 1 \leq k \leq n. \tag{2}$$

The uncertain dataset $UD = \{x_1, x_2, \ldots, x_n\}$ generates a possible world set $pw = \{x_1', x_2', x_3', \ldots, x_N'\}$, composed of all possible instances from $UD$. The sum of probability values of all possible worlds is 1 (*Liu, Zhang & Zhang, 2019*):

$$\sum_{v=1}^{N} p(w_v) = 1. \tag{3}$$

The possible world set is generated based on a uniform probability distribution (*Liu, Zhang & Zhang, 2019*).

**Table 1 Probability distribution of symptoms for migraine.**

| Symptom | Probability |
|---|---|
| Nausea | $p(S_1) = 0.333$ |
| Severe pain around the left eye | $p(S_2) = 0.333$ |
| Headache | $p(S_3) = 0.333$ |

### A MayBMS repair key

The MayBMS Probabilistic database management system incorporates the Repair Key operator to generate possible worlds within the probabilistic framework (*Bekkers, 2022*). This operator assigns a probability value to each record based on a uniform probability distribution if weights are not provided. For $n$ records with the same identifier, the operator assigns a probability of $1/n$ to each record (*Bekkers, 2022*).

In the context of the disease migraine, three symptoms are considered, each with equal probability values indicating the weight of each symptom in the disease. Each record, denoted as $S_i$, is associated with a probability value $p(S_i)$. The disease-symptoms information is represented by a discrete set $S_1, S_2, \ldots, S_i$, consisting of $m \in \mathbb{N}$ possible worlds. It is crucial to ensure that the sum of probability values assigned to all records is at most one, aligning with the probabilistic framework's requirements (*Züfle, 2021*).

$$\sum_{i=1}^{m} p(s_i) \leq 1. \tag{4}$$

The probability distribution of symptoms for migraine, calculated using the MayBMS Repair Key operator, is presented in Table 1.

## LITERATURE REVIEW

Researchers are developing novel clustering algorithms, specifically partition-based, density-based, and possible world-based, to effectively handle the challenges posed by uncertain data (*Liu et al., 2021*). In the realm of partition-based algorithms, the K-means algorithm has been extended to introduce UK-means, specifically designed to handle uncertain data (*Chau et al., 2006*; *Li et al., 2020*). While UK-means generates spherical clusters, it lacks the ability to produce clusters of arbitrary shapes (*Liao & Liu, 2016*). To address this limitation, researchers optimized UK-means by introducing CK-means, which utilizes the moment of inertia of rigid bodies (*Lee, Kao & Cheng, 2007*). Additionally, a modification called DUK-means was developed to enable UK-means to operate in distributed network environments (*Zhou et al., 2018*). Another extension, UK-medoids, incorporates uncertain distances into the K-medoids algorithm (*Gullo, Ponti & Tagarelli, 2008*).

Density-based clustering algorithms offer an alternative approach. Notably, the FDBSCAN algorithm modifies the traditional density-based clustering method, DBSCAN (*Kriegel & Pfeifle, 2005a*). In FDBSCAN, the distance distribution function serves as a similarity measure. Similarly, the FOPTICS algorithm extends clustering for uncertain

data by leveraging the OPTICS algorithm and a probabilistic definition (*Kriegel & Pfeifle, 2005b*). However, *Zhang, Liu & Zhang (2017)* identified drawbacks in both FDBSCAN and FOPTICS, including the loss of uncertain information, high time complexity, and non-adaptive thresholds. To overcome the limitations arising from the unrealistic independent distance assumption, researchers have explored possible world model-based algorithms as an alternative to partition-based and density-based approaches. In a study by *Volk et al. (2009)*, an algorithm conducts clustering independently on each possible world and subsequently integrates the clustering results into a final outcome. Similarly, *Züfle et al. (2014)* adopts a similar strategy, performing clustering on individual possible worlds and selecting the representative clustering result as the final outcome. In a different work, *Liu, Zhang & Zhang (2018)* introduced an algorithm that learns a consensus affinity matrix for clustering uncertain data by constructing independent affinity matrices for each possible world. By leveraging the consistency principle, the authors aim to maximize the benefits of clustering uncertain data. However, this approach lacks support for updating the affinity matrix of each possible world, resulting in a diminished capacity for consistency learning (*Liu et al., 2019*). Consequently, the efficiency of the consistency principle is compromised. Furthermore, the marginal possible worlds pose challenges as they violate the learning principle and detrimentally impact the results by being treated equally. To mitigate this issue, the authors in *Liu et al. (2019)* propose a post-processing heuristic technique that selectively retains true worlds while filtering out marginal possible worlds due to their limited representational capability. Nevertheless, criticisms from research articles (*Sharma & Seal, 2021a*; *Li et al., 2021*) highlight concerns regarding the time-consuming and inefficient nature of the proposed post-processing heuristic technique, which could compromise the effectiveness of the consistency principle and the overall performance of the clustering process.

Furthermore, traditional clustering techniques like K-means have limitations due to their exclusive assignment of objects to single clusters. Overlapping clustering algorithms have been proposed as alternatives. One key algorithm is the Overlapping K-means (OKM) algorithm, which extends the conventional K-means algorithm (*Cleuziou, 2007*). Numerous OKM extensions tackle various facets of overlapping clustering, including OKMED, WOKM, KOKM/KOKM$\phi$, and parametrized OKM methods (*Khanmohammadi, Adibeig & Shanehbandy, 2017*). However, KHM-OKM can struggle with complex overlapping patterns and may perform poorly if initialization fails to identify representative cluster centers. The Fuzzy C-Means (FCM) algorithm assigns objects to clusters based on membership values between 0 and 1 (*Bezdek, Ehrlich & Full, 1984*). However, FCM is unsuitable for clustering overlapping symptoms in our dataset because it assumes continuous data, struggles with categorical data, and relies on Euclidean distance. It fails to account for uncertainties and interactions in overlapping symptoms, making it inadequate for capturing complex relationships across multiple diseases. The algorithm discussed in *Askari (2021)* revises the traditional FCM algorithm by using adaptive exponential functions to mitigate noise and outliers and modifies constraints to prevent large clusters from attracting the centers of smaller clusters. However, it is not suitable for clustering overlapping symptoms as it does not address the complexities of overlapping

symptoms and inherent uncertainty in medical diagnosis. Similarly, the method proposed by *Yu et al. (2020)*, "An active three-way clustering method *via* low-rank matrices for multi-view data," enhances clustering accuracy through low-rank matrix representations and active learning to refine clusters iteratively. Despite its advancements, this approach is also not suitable for clustering overlapping symptoms in medical diagnosis because it focuses on high-dimensional multi-view data and does not adequately address the inherent uncertainties and categorical nature of overlapping symptoms.

Clustering algorithms have evolved to meet specific domain needs. While many similarity measures exist for numeric data (*Chai et al., 2021*; *Oyewole & Thopil, 2023*; *Amer & Abdalla, 2020*), few cater to categorical data (*Kar, Mishra & Mohanty, 2023*), posing a challenge for datasets containing symptom names (*Šulc & Řezanková, 2019*; *Dinh & Huynh, 2020*). Traditional measures, grouping data by proximity, are unsuitable for overlapping symptom clusters (*Berbague et al., 2021*). Numeric similarity measures like Euclidean and Manhattan distances are inadequate for nominal categorical data (*Kar, Mishra & Mohanty, 2023*; *Šulc & Řezanková, 2019*; *Dinh & Huynh, 2020*), and may fail to capture correlations among data objects with the same probability distribution (*Sharma & Seal, 2021a*). Hamming distance is a common similarity measure for categorical data (*Esposito et al., 2000*), but measures like Jaccard, Sokal–Michener, and Grower–Legendre are limited for nominal data without inherent ordering (*Mumtaz & Giese, 2020*). Frequency-based measures utilize attribute frequency distributions to overcome these limitations but may perform less efficiently on complex datasets with domain dependencies (*Mumtaz & Giese, 2020*). Categorical data similarity measurement is more complex than numeric data due to its qualitative nature (*Han, Pei & Kamber, 2012*). New probability-based measures are needed for handling uncertain probabilistic data. Traditional numeric data distance functions are unsuitable for categorical data, highlighting the necessity for specialized approaches. This need is addressed in the work of Bekkers, which discusses leveraging probabilistic databases for modeling and simulating infectious diseases (*Bekkers, 2022*).

In conclusion, while significant advancements have been made in clustering algorithms to handle uncertain data, challenges remain in effectively clustering overlapping symptoms, particularly in medical diagnosis. The proposed approach in this research aims to fill this gap by introducing a novel clustering algorithm that considers the precise weight of each symptom in every disease, facilitating the generation of overlapping clusters that accurately represent symptom associations within the context of different diseases. This innovative approach not only addresses the limitations of existing clustering algorithms but also enhances the accuracy and reliability of disease diagnosis, ultimately contributing to better patient outcomes and more effective treatment strategies.

## MATERIALS AND METHODS

First, the raw dataset is pre-processed to ensure its suitability for analysis, as machine learning algorithms typically require properly formatted data. Various pre-processing techniques are employed to transform the raw data into the appropriate format. Next, probability values are calculated to determine the significance of each symptom in relation

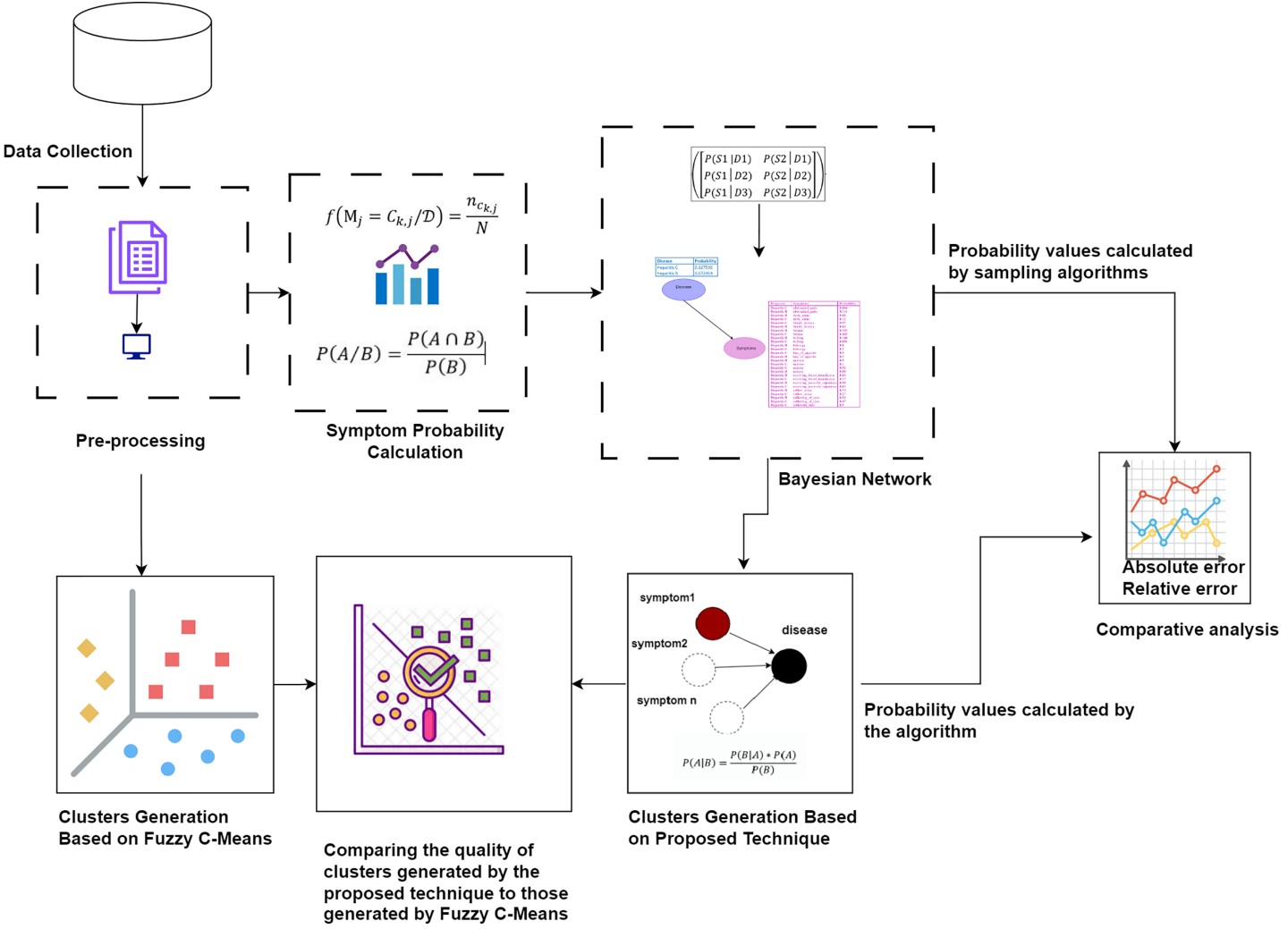

**Figure 1 The proposed methodology for clustering uncertain overlapping symptoms of multiple diseases.** The process includes data collection, pre-processing, symptom probability calculation, and clustering using Bayesian Network and Naïve Bayes Classifier. Comparative analysis of the clusters generated by the proposed technique and Fuzzy C-Means is performed, utilizing absolute and relative error metrics for evaluation. Figure Components Sources: Methodology diagram template: Created using Draw.io. The pages and computer in "Pre-processing" shapes: Draw.io. The graph in "Symptom Probability Calculation": Draw.io. Relational database in "Bayesian Network": Created using MS Paint. Absolute error, Relative error chart: IconFinder. Magnifying glass over data points: Flaticon. Clusters Generation Based on Fuzzy C-Means: Flaticon. Icon source credits: Flaticon.com.

to diseases. Bayesian network and naïve Bayes classifier are utilized to cluster the symptoms by generating possible worlds based on the calculated probabilities. This approach allows for a detailed analysis of the probabilistic relationships between symptoms and diseases, identifying patterns and associations that can improve the accuracy of disease diagnosis. The probability values calculated by our proposed technique are compared with the probability values calculated by sampling algorithms through absolute and relative errors. The quality of the clusters generated by the proposed technique is compared to those generated by Fuzzy C-Means. This methodology is visually represented in Fig. 1.

---

**Algorithm 1 Calculate probability of symptoms.**

**Require:** Database with patient, disease, and symptom information

**Ensure:** Probability of Symptoms for each disease

1: **function** CalculateProbability Database

2:      $D \leftarrow$ Database

3:      $D_{\text{diseaseIDs}} \leftarrow \{d.\text{DiseaseID} \mid d \in D.\text{Disease}\}$                    ▷ Fetch disease IDs

4:      $D_{\text{patientIDs}} \leftarrow \{p.\text{PatientID} \mid p \in D.\text{Patient} \land \exists d \in D.\text{Disease} : d.\text{DiseaseID} \in D_{\text{diseaseIDs}} \land$
     $p.\text{DiseaseID} = d.\text{DiseaseID}\}$                    ▷ Group patients by disease

5:      $D_{\text{symptomIDs}} \leftarrow \{s.\text{SymptomID} \mid s \in D.\text{Symptom} \land \exists p \in D.\text{Patient} : p.\text{PatientID} \in D_{\text{patientIDs}} \land$
     $s.\text{PatientID} = p.\text{PatientID}\}$              ▷ Fetch symptoms for each disease group

6:      $D_{\text{symptomCounts}} \leftarrow \{\}$                    ▷ Count occurrence of each symptom in a disease

7:      $n \leftarrow \text{length}(D_{\text{symptomIDs}})$

8:      $N \leftarrow n$

9:      $D_{\text{symptomProbability}} \leftarrow \{\}$                 ▷ Probability of each symptom in a disease

10:     **for** $i = 1$ to $n$ **do**

11:          $c \leftarrow 1$

12:          **for** $j = i + 1$ to $n$ **do**

13:              **if** $D_{\text{symptomIDs}}[i] = D_{\text{symptomIDs}}[j]$ **then**

14:                  $c \leftarrow c + 1$

15:                  $D_{\text{symptomIDs}}[j] \leftarrow -1$

16:              **end if**

17:          **end for**

18:          **if** $D_{\text{symptomIDs}}[i] \neq -1$ **then**

19:              $D_{\text{symptomCounts}}[D_{\text{symptomIDs}}[i]] \leftarrow c$

20:          **end if**

21:     **end for**

22:     **for** $i = 1$ to $N$ **do**

23:          **if** $D_{\text{symptomCounts}}[i] \neq -1$ **then**

24:             $D_{\text{symptomProbability}}[i] \leftarrow \dfrac{D_{\text{symptomCounts}}[i]}{N}$

25:          **end if**

26:     **end for**

27:     **return** $D_{\text{symptomProbability}}$

28: **end function**

---

## Dataset

We processed a dataset using preprocessing techniques to prepare raw data for our algorithm, aiming to identify symptom relationships through symptom clusters. We utilized a Kaggle dataset (*Kaushil, 2020*) consisting of two CSV files for training and testing purposes. This dataset comprises 133 columns, mapping symptoms to 42 diseases with binary associations.

## Calculating probability values

We employed the relative frequency approach to analyze symptom datasets, facilitating the creation of overlapping clusters. This method calculates probability values of symptoms, informing a clustering technique for identifying overlapping clusters (*Fazzolari et al., 2021*). By applying this technique, we unveil meaningful relationships between symptoms. To address data uncertainty, we propose an algorithm outlined in Algorithm 1. This algorithm employs the relative frequency formula:

$$S_{k,j} = \frac{n_{S_{k,j}}}{N_{d,j}}. \tag{5}$$

Here, $S_{k,j}$ represents the relative frequency of the $k$th symptom in disease $D_j$, calculated by dividing the number of occurrences of that symptom within the disease $n_{S_{k,j}}$ by the total number of symptoms $N_{d,j}$ observed in patients diagnosed with disease $D_j$. This statistical approach ensures robust estimation of symptom weights.

The integration of the relative frequency approach and the proposed algorithm enables comprehensive analysis, yielding insights crucial for disease characterization, diagnosis, and treatment strategies in medical research and clinical practice.

## Conditional probability matrix

The conditional probability matrix (CPM) is a structured matrix that represents the conditional probabilities between two discrete random variables, $X$ and $Y$. Let $P$ be the probability function on the space $D$, where $X$ takes values in $\{1, 2, \ldots, m\}$ and $Y$ takes values in $\{1, 2, \ldots, n\}$.

The CPM, given in Eq. (6), is defined as follows (*Staic, 2022*):

$$\beta = (b_{u,v}^t). \tag{6}$$

Here, $\beta = (b_{u,v}^t)$ represents the conditional probability. Each entry in the CPM represents the conditional probability of $Y$ being equal to $t$ given that $X$ is between $u$ and $v$, i.e.,

$$b_{u,v}^t = P(Y = t | u < X \leq v). \tag{7}$$

The calculation of the conditional probability is given by the formula:

$$P(Y = t | u < X \leq v) = \frac{P(Y = t, u < X \leq v)}{P(u < X \leq v)} \tag{8}$$

where $1 \leq u < v \leq m$ and $1 \leq t \leq n$.

Columns in the matrix correspond to the variable (X), while rows represent specific states of the conditioning variable (Y). Each entry in the matrix reflects the conditional probability of an event (X) occurring given that another event (Y) has already happened. This organization allows for the analysis and understanding of the relationships between variables. For instance, in the case of a symptom (S) conditioned by a disease (D), the CPM will store the conditional probabilities, providing insights into the relationship between the

**Table 2 Conditional probabilities of symptoms given diseases.**

| $(u, v)$ | Probability |
|---|---|
| 1 | $P(Y = 1\|u \leq X \leq v) = 0.4$ |
| 2 | $P(Y = 1\|u \leq X \leq v) = 0.75$ |
| 3 | $P(Y = 1\|u \leq X \leq v) = 0.9$ |
| 4 | $P(Y = 1\|u \leq X \leq v) = 0.3$ |
| 5 | $P(Y = 1\|u \leq X \leq v) = 0.9$ |
| 1 | $P(Y = 2\|u \leq X \leq v) = 0.6$ |
| 2 | $P(Y = 2\|u \leq X \leq v) = 0.25$ |
| 3 | $P(Y = 2\|u \leq X \leq v) = 0.1$ |
| 4 | $P(Y = 2\|u \leq X \leq v) = 0.7$ |
| 5 | $P(Y = 2\|u \leq X \leq v) = 0.1$ |

disease and the symptom. Overall, the CPM serves as a valuable tool for quantifying and investigating the conditional probabilities, facilitating the analysis and interpretation of the relationships between variables.

$$CPM = \begin{bmatrix} P(S1|D1) & P(S2|D1) & P(S3|D1) \\ P(S1|D2) & P(S2|D2) & P(S3|D2) \\ P(S1|D3) & P(S2|D3) & P(S3|D3) \end{bmatrix}.$$

*Example 1: The problem at hand involves determining the weights of different symptoms in different diseases. To address this, we utilize the conditional probability $P(Y = t|u \leq X \leq v)$, where X represents symptoms (S) and Y represents diseases (D). In our scenario, the set of all symptoms stored in the database is denoted as $X : S \rightarrow \{1, 2, 3, 4, 5\}$. Here, if $X = 1$, it corresponds to the symptom headache, $X = 2$ represents fever, and so on. Similarly, the set of all diseases stored in the database is denoted as $Y : D \rightarrow \{1, 2\}$, with the convention that $Y = 1$ corresponds to Dengue, and $Y = 2$ corresponds to Typhoid. By utilizing conditional probability, we can analyze the relationships between symptoms (X) and diseases (Y) and estimate the weights of symptoms in different diseases as given in the Table 2.*

## Bayesian network

A Bayesian network (BN) is a graphical model where nodes represent random variables, and edges depict their relationships (*Kyrimi et al., 2020*). Each BN child node is associated with a conditional probability table (CPT) specifying its distribution. For instance, in the BN depicted in Fig. 2, the disease node serves as the parent node of the symptom node, indicating a direct impact of disease on the symptom. Probabilistic reasoning is performed by considering all the nodes, edges, and CPTs. The initial given probability values are known as prior probabilities.

Inference within the BN is conducted using Eq. (10), which involves calculating probabilities of diseases given specific symptoms, such as

$P(\text{disease} = \text{Hepatitis B} \mid \text{Hepatitis C}(\text{symptoms} = \text{"itching"}))$.

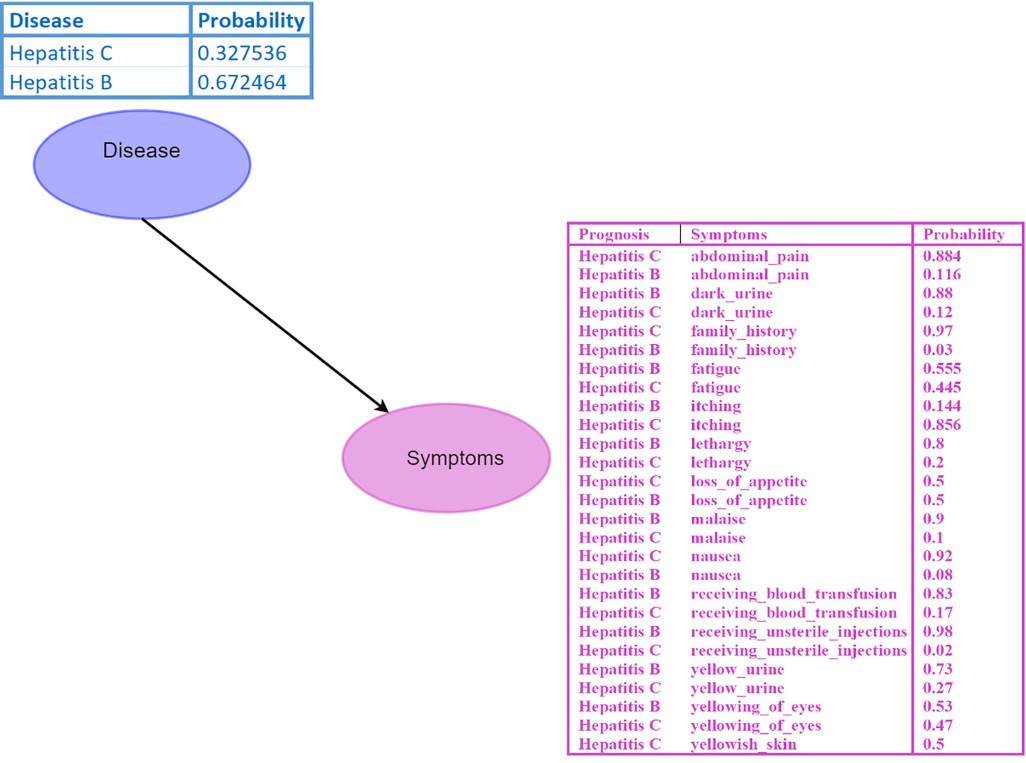

**Figure 2 Bayesian network (BN).** The "Disease" node acts as the parent node to the "Symptoms" node, which contains all symptoms and their conditional probability tables (CPTs). The directed edge represents the influence of the disease on the symptoms. Figure components sources: Relational database in "Bayesian Network": Created using MS Paint.

The BN topology is represented as a set of vertices (*V*) and edges (*E*) (*Trösser, de Givry & Katsirelos, 2021*).

$$BN = \langle V, E \rangle. \tag{9}$$

Here, *V* represents the collection of all nodes, and *E* represents the set of directed edges that indicate the logical relationships between nodes. For example, a directed edge $\langle n_j, n_i \rangle$ signifies that $n_i$ is the child node and $n_j$ is the parent node of $n_i$, denoted as $\omega(n_i)$ (*Kitson et al., 2023*).

The joint probability distribution in the BN, considering a set of discrete nodes $V = \{n_1, n_2, \ldots, n_p\}$, is given by Eq. (10):

$$P(n_i, \ldots, n_j) = \prod_{i=1}^{j} P(n_i | \omega(n_i)). \tag{10}$$

While Bayesian Networks are effective in managing uncertainty, determining the conditional probability tables can be challenging and computationally demanding. Furthermore, the size of the CPT grows exponentially as the number of parent nodes increases (*Bibartiu et al., 2024*).

### Probabilistic inferences

Probabilistic inference in a Bayesian network determines the posterior probability distribution for nodes based on evidence from other nodes, allowing for belief updating (*Rouigueb et al., 2023*). There are two main classes of algorithms: exact and approximate methods (*Spallitta et al., 2024*).

Inference in simple tree structures can use local computations and message passing (*Wang, AmrilJaharadak & Xiao, 2020*). However, when nodes are connected by multiple paths, inference becomes more complex. Exact inference may become computationally infeasible, requiring approximate inference algorithms. Both methods are NP-hard (*Behjati & Beigy, 2020*).

### Exact inference in chains in two node network

Suppose we aim to determine the causal effect of yellowing of eyes on hepatitis B. Given a prior probability of yellowing of eyes, $P(\text{Yellowing of Eyes} = \text{T}) = 0.110145$, and conditional probabilities for hepatitis B based on the presence or absence of yellowing, $P(\text{Hepatitis B} = \text{T}|\text{Yellowing of Eyes} = \text{T}) = 0.53$ and $P(\text{Hepatitis B} = \text{T}|\text{Yellowing of Eyes} = \text{F}) = 0.47$, we proceed with the following reasoning diagnostic:

$\text{Bel}(\text{Yellowing of Eyes} = \text{T}) = \beta \times 0.05837685$
$\text{Bel}(\text{Yellowing of Eyes} = \text{F}) = \beta \times 0.05176815$

where $\beta = \frac{1}{0.05837685 + 0.05176815}$, derived from the constraint that the sum of beliefs equals 1. Thus, we update the beliefs as:

$$\text{Bel}(\text{Yellowing of Eyes} = \text{T}) = \frac{0.05837685}{0.05837685 + 0.05176815} = 0.53$$

$$\text{Bel}(\text{Yellowing of Eyes} = \text{F}) = \frac{0.05176815}{0.05837685 + 0.05176815} = 0.47$$

## Proposed technique based on Bayesian network and naive Bayes algorithm

In real-time medical scenarios, patients often experience multiple symptoms rather than just a single symptom during their illness. Furthermore, diseases can cause multiple symptoms, and many diseases share overlapping symptoms. Hence, it becomes crucial to develop methods for obtaining probabilistic inferences of diseases based on symptoms. To address this, we can utilize the Bayes theorem in a modified form (*Korb & Nicholson, 2010*):

$$P(D|S) = \frac{P(S|D)P(D)}{P(S)}. \tag{11}$$

Here, the variable $D$ represents the cluster variable, which corresponds to diseases, while the variable $S$ represents the symptom parameters or features, denoted as

$S = S_1, S_2, S_3, \ldots, S_n$. By substituting $S$ and expanding Eq. (11) using the Chain Rule, we obtain (*Chen et al., 2020*):

$$P(D|S_1, S_2, S_3, \ldots, S_n) = \frac{P(S_1|D)P(S_2|D)P(S_3|D)\ldots P(S_n|D)P(D)}{P(S_1)P(S_2)P(S_3)\ldots P(S_n)}. \tag{12}$$

For all diseases, the denominator remains the same. Therefore, the denominator can be removed, and we can introduce proportionality (*Hosein & Baboolal, 2022*):

$$P(D|S_1, S_2, S_3, \ldots, S_n) \approx P(D) \prod_{i=1}^{n} (S_i|D). \tag{13}$$

In Eq. (13), the disease variable is multivariate, representing different disease clusters with distinct sets of symptoms. To determine the maximum probability for a specific disease cluster, we can calculate (*Chen et al., 2020*; *Hosein & Baboolal, 2022*):

$$\arg\max_{D} \left( P(D) \prod_{i=1}^{n} (S_i|D) \right). \tag{14}$$

By employing Eq. (13), we can estimate the probabilities of different disease clusters based on observed symptoms, enabling effective disease diagnosis and treatment decisions.

*Example: 2 Consider the probability calculation for Hepatitis B and Hepatitis C. We denote the diseases as $D = d_1, d_2$, with $d_1$ as Hepatitis B and $d_2$ as Hepatitis C. Probability values are $P(d_1) = 0.672464$ and $P(d_2) = 0.327536$. Symptom set: $S = s_1, s_2, s_3, s_4, s_5$, where $s_1$ (abdominal pain), $s_2$ (dark urine), $s_3$ (itching), $s_4$ (fever), and $s_5$ (fatigue). Conditional probabilities are given as follows: $P(s_1|d_1) = 0.116$, $P(s_2|d_1) = 0.880$, $P(s_3|d_1) = 0.144$, $P(s_4|d_1) = 0.708$, $P(s_5|d_1) = 0.555$, $P(s_1|d_2) = 0.884$, $P(s_2|d_2) = 0.120$, $P(s_3|d_2) = 0.856$, $P(s_4|d_2) = 0.292$, $P(s_5|d_2) = 0.445$. Using the Naïve Bayes classifier, we assume that the symptoms are conditionally independent given the disease.*

Therefore, we can calculate the probabilities using the following formula given in Eq. (13):

$$P(D|S_1, S_2, S_3, \ldots, S_n) \approx P(D) \prod_{i=1}^{n} (S_i|D).$$

Now, let us recalculate the probabilities for Hepatitis B and Hepatitis C based on the given symptoms:

$$P(D = \text{Hepatitis B}|S_{\text{abdominal pain}}, S_{\text{dark urine}}, S_{\text{itching}}, S_{\text{fever}}, S_{\text{fatigue}})$$
$$\approx (0.672464)[(0.116 \times 0.880 \times 0.144 \times 0.708 \times 0.555)]$$
$$\approx 0.00388417182691$$

$$P(D = \text{Hepatitis C}|S_{\text{abdominal pain}}, S_{\text{dark urine}}, S_{\text{itching}}, S_{\text{fever}}, S_{\text{fatigue}})$$
$$\approx (0.327536)[(0.884 \times 0.120 \times 0.856 \times 0.292 \times 0.445)]$$
$$\approx 0.003864641196796723$$

### Normalization

Normalization is crucial in probability theory to confine probabilities within the range of 0 to 1, representing event likelihood. This ensures that probabilities sum up to 1, forming a valid probability distribution.

Normalization involves dividing each probability by the sum of all probabilities, yielding the Eq. (15) (*Li et al., 2022*):

$$\text{Normalized Probability} = \frac{\text{Probability}}{\sum \text{Probabilities}}. \tag{15}$$

Here, Normalized Probability represents the normalized probability, and Probability is the original probability of an event. The denominator, $\sum$ Probabilities, signifies the sum of all probability values.

Normalization is a fundamental technique for creating valid probability distributions, enhancing result reliability, and facilitating interpretation across diverse fields, including probability theory, machine learning, and statistics.

As an example, let us consider the normalization of probabilities for two diseases, Hepatitis B and Hepatitis C:

Now, let us calculate the normalized probabilities for Hepatitis B and Hepatitis C:

$$\text{Normalized Probability for Hepatitis B} = \frac{0.00388417182691}{0.00388417182691 + 0.003864641196796723}$$
$$\approx 0.5013$$

$$\text{Normalized Probability for Hepatitis C} = \frac{0.003864641196796723}{0.003864641196796723 + 0.00388417182691}$$
$$\approx 0.4987$$

### Output of the proposed technique

Following normalization, we observe that the probability for Hepatitis B is now approximately 0.5013, while the probability for Hepatitis C is also around 0.4987. This outcome aligns with expectations and ensures that the probabilities sum up to 1, reflecting a valid and reliable probability distribution suitable for disease clustering or classification tasks.

Using naive Bayes algorithms, the clusters will be created as shown in Fig. 3.

Figure 4 illustrates how different symptoms group together for multiple diseases. Each color represents a different cluster of symptoms, indicating their association with specific diseases.

## Inference techniques for Bayesian networks

### Rejection sampling

Rejection sampling is a fundamental technique used for computing conditional probabilities in Bayesian networks by generating samples and rejecting those that do not align with the given evidence (*Kwisthout, 2018*). A Bayesian network comprises a set of

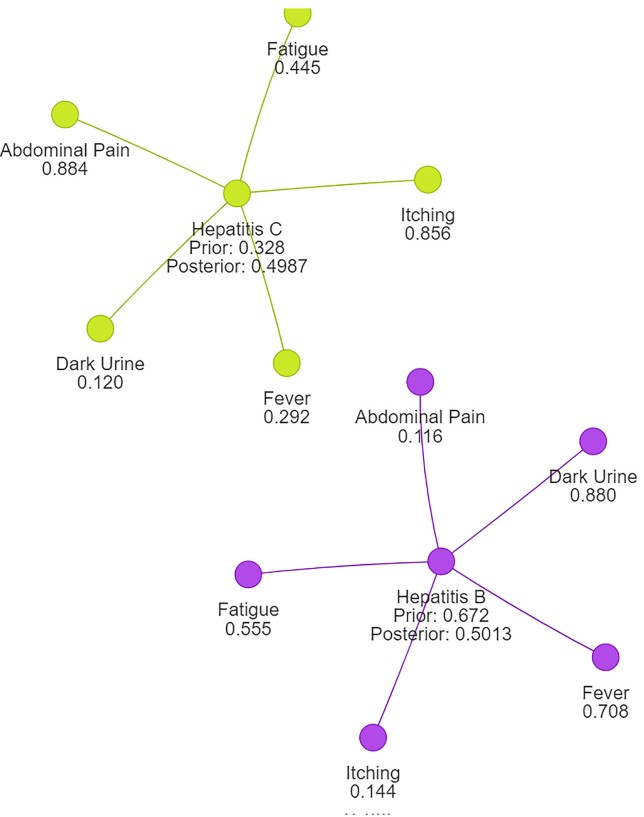

**Figure 3 Symptom clusters for Hepatitis B and Hepatitis C using a naive Bayes algorithm, showing the overlap and associations between symptoms based on conditional probabilities.**

nodes $\{n_1, n_2, \ldots, n_j\}$, where each node $n_i$ is associated with a Conditional Probability Table (CPT) denoted as $P(n_i|\omega(n_i))$, where $\omega(n_i)$ represents the parents of $n_i$.

During probabilistic inference, our primary goal is to determine the conditional probability $Pr[N_q|N_e]$ for specific random variables, where $q$ represents the query variable and $e$ represents the evidence, with $q \neq e$. To achieve this, we generate $X$ samples from the joint distribution of the Bayesian Network, denoted as $Pr[N_1, \ldots, N_j]$, represented by $\{(n_1^{(i)}, \ldots, n_j^{(i)})\}_{i=1}^{X}$. The probability is calculated using the following Eq. (10):

$$P(n_i, \ldots, n_j) = \prod_{i=1}^{j} P(n_i|\omega(n_i)).$$

To compute the probability for the query based on the given evidence, we utilize the CPT as follows:

$$\theta(n_q, n_e) = \frac{\sum_{i=1}^{X} 1[n_q^{(i)} = n_q] \cdot 1[n_e^{(i)} = n_e]}{\sum_{i=1}^{X} [n_e^{(i)} = n_e]}. \tag{16}$$

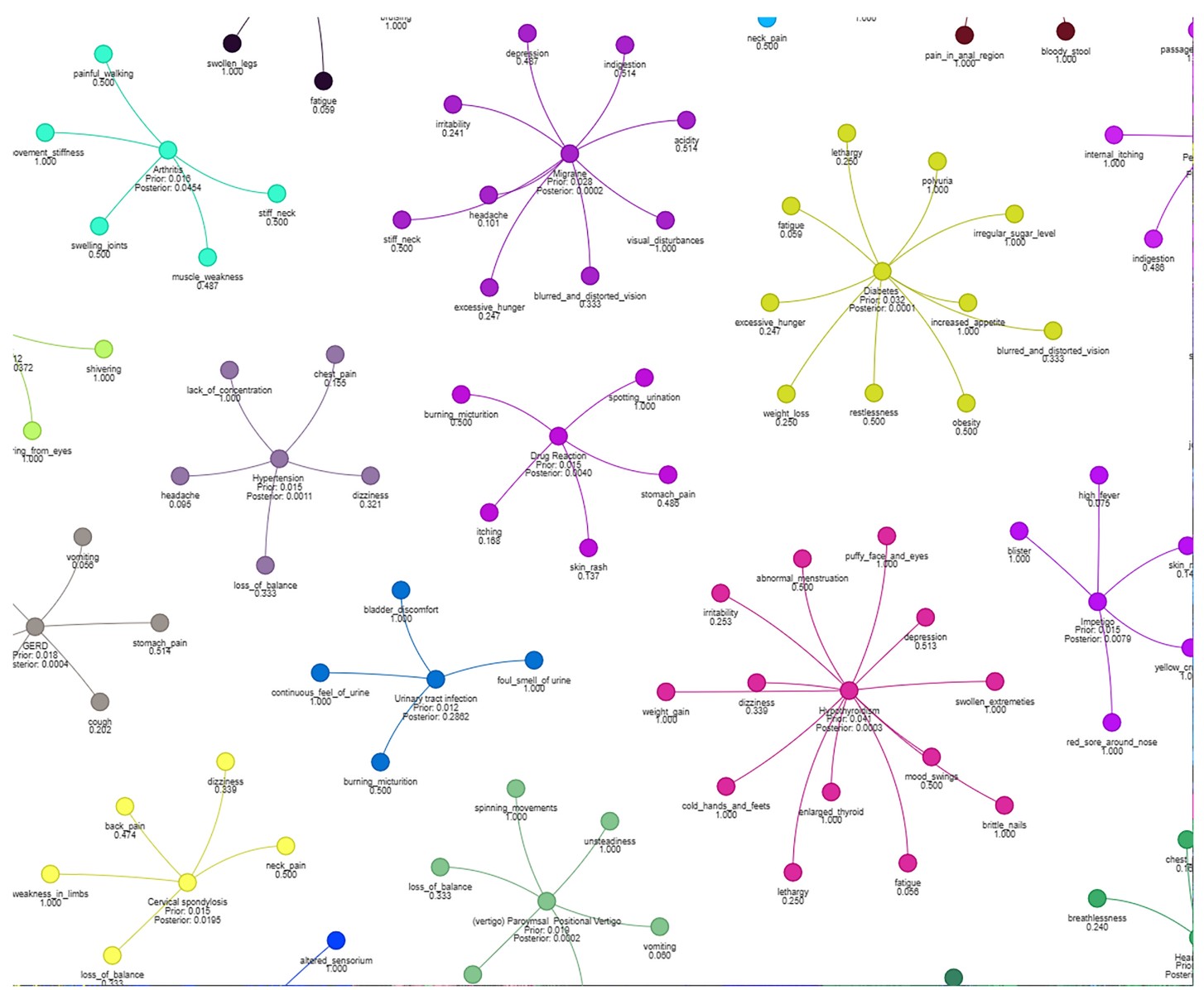

**Figure 4  Results of a clustering algorithm applied to a dataset of symptoms.**

Here, $(n_q, n_e)$ represent the possible outcomes for the random variables $(N_q, N_e)$, where $n_q$ and $n_e \in \{0, 1\}$ for binary nodes. An approximation of the conditional probability is given by:

$$Pr[N_q = n_q | N_e = n_e] \approx \theta(n_q, n_e).$$
(17)

When $X$ is large, the above equation can be rewritten as:

$$\lim_{X \to \infty} \theta(n_q, n_e) = Pr[N_q = n_q | N_e = n_e].$$
(18)

Furthermore, the relationship between $\theta(n_q, n_e)$ and the number of samples can be defined as:

$$\theta(n_q, n_e) = \frac{J(n_q, n_e)}{J(n_e)}. \tag{19}$$

Here, $J(n_q, n_e)$ represents the number of samples in which the random variables $N_q$ and $N_e$ take the values $n_q$ and $n_e$, respectively. $J(n_e)$ denotes the number of samples in which the random variable $N_e$ takes the value $n_e$. This relationship can be further expanded as:

$$\theta(n_q, n_e) = \frac{J(n_q, n_e)}{X \cdot X / J(n_e)}. \tag{20}$$

It is important to note the following relationship between conditional probabilities and joint probabilities:

$$Pr[N_q = n_q | N_e = n_e] = \frac{Pr[N_q = N_q, N_e = n_e]}{Pr[N_e = n_e]}. \tag{21}$$

As $X$ becomes sufficiently large, we approximate:

$$Pr[N_q = N_q, N_e = n_e] \approx \frac{J(n_q, n_e)}{X} \tag{22}$$

and,

$$Pr[N_e = n_e] = \frac{J(n_e)}{X}. \tag{23}$$

Consequently, we obtain:

$$Pr[N_q = n_q | N_e = n_e] = \theta(n_q, n_e). \tag{24}$$

Samples that do not satisfy the condition $n_e^{(i)} \neq n_e$ are rejected as they do not contribute to either the numerator or the denominator of $\theta$. This rejection of samples that do not align with the evidence is a key characteristic of rejection sampling.

### Likelihood weighting

Let $N_{(-e)}$ represent the set of all variables in the network except for $N_e$, given by $N_{(-e)} = N_1, \ldots, N_{(e-1)}, N_{(e+1)}, \ldots, N_j$. Similarly, let $n_{(-e)}^{(i)}$ represent the values of the variables in $N_{(-e)}$ for the $i$th sample, denoted as $n_{(-e)}^{(i)} = (n_1^{(i)}, \ldots, n_{(e-1)}^{(i)}, n_{(e+1)}^{(i)}, \ldots, n_j^{(i)})$.

To perform likelihood weighting, we generate $X$ samples from the Bayesian Network BN. For each sample, we assign values to the nodes $N_i$ based on their respective Conditional Probability Tables (CPTs) (*Yuan & Druzdzel, 2006*). For example, if $N_1$ is the root node, we sample $n_1$ from $Pr[N_1]$, and then sample $n_2$ from $Pr[N_2 | N_1 = n_1]$, and so on.

The probability is calculated using the following formula:

$$\theta(n_q, n_e) = \frac{\sum_{i=1}^{X} 1[n_q^{(i)} = n_q] \cdot Pr[N_e = n_e | (\omega(N_e))^{(i)}]}{\sum_{i=1}^{X} Pr[N_e = n_e | (\omega(N_e))^{(i)}]}.$$ (25)

In this equation, $(n_q, n_e)$ represents the possible outcomes for the random variables $(N_q, N_e)$. The numerator involves summing over the $X$ samples, considering only the samples where the value of $n_q^{(i)}$ matches with $n_q$. The indicator function, denoted as $1[n_q^{(i)} = n_q]$, is a binary function that evaluates to 1 if $N_q$ in the $i$th sample matches $n_q$, and 0 otherwise. This ensures that we only consider the samples that are consistent with the desired value of $N_q$.

For each sample, the numerator multiplies the indicator function by the conditional probability $Pr[N_e = n_e | (\omega(N_e))^{(i)}]$. This conditional probability represents the likelihood of the evidence variable $N_e$ being equal to $n_e$, given the values of its parents $(\omega(N_e))$ in the $i$th sample. This product captures the contribution of each sample that satisfies both the value of $N_q$ and $N_e$.

The denominator sums over all $X$ samples and considers the conditional probabilities $Pr[N_e = n_e | (\omega(N_e))^{(i)}]$ for each sample. This sums up the likelihood of the evidence variable $N_e$ being equal to $n_e$, given the values of its parents, across all samples.

### Gibbs sampling

In a Bayesian network, Gibbs sampling is employed to generate $X$ samples. The samples are denoted as $\{(n_1^{(i)}, \ldots, n_{(e-1)}^{(i)}, n_{(e+1)}^{(i)}, \ldots, n_j^{(i)})\}^X$. Each sample represents a configuration where $n_j^{(i)}$ corresponds to the value of variable $N_j$ in the $i$th sample. Gibbs sampling proceeds by iteratively generating a new sample. To generate a new sample, we assign a new value $n_j^{(i+1)}$ to each variable $N_j$. This assignment is based on the conditional probability distribution, conditioned on the remaining variables and evidence, which is expressed as:

$$n_j^{(i+1)} \leftarrow P\left(n_j | n_{(-j)}^{(i)}, e\right)$$ (26)

Here, $n_{(-j)}$ represents all variables except $n_j$. The conditional probability $P(n_j | n_{(-j)}^{(i)}, e)$ is used to determine the probability of the new value $n_j^{(i+1)}$ for variable $N_j$, given the values of the other variables in $n_{(-j)}^{(i)}$ and any observed evidence $e$.

The probability distribution for each variable $N_j$, conditioned on the evidence $e$, can be calculated using the equation (*Bidyuk & Dechter, 2002*):

$$P(n_j | e) = \frac{1}{X} \sum_{i=1}^{X} P(n_j | n_{(-j)}).$$ (27)

In this Eq. (27), the probability $P(n_j | n_{(-j)})$ represents the probability of variable $N_j$ taking the value $n_j$, given the values of the other variables in $n_{(-j)}$. The sum over $X$ samples

ensures that we consider the distribution of variable $N_j$ across all the generated samples, with each sample contributing equally $(1/X)$.

By iteratively updating the values of variables using conditional probabilities and generating new samples, Gibbs sampling provides an approximation of the joint probability distribution of the variables in the Bayesian network, allowing for probabilistic inference.

## Experimental setup

We implemented our clustering algorithm using Python 3.8 with libraries such as pandas, numpy, matplotlib, networkx, pgmpy, and pyvis. Experiments were conducted on a Windows 10 Pro system with an Intel Core i7-8550U CPU @ 1.80 GHz and 20 GB of RAM.

The BayesianEstimator used a "BDeu" prior with an equivalent sample size of 10 on normalized symptom data. Likelihood Weighting, Rejection Sampling, and Gibbs Sampling were employed with sample sizes from $10^{2.5}$ to $10^5$ across 10 experiments per size, typically converging within 300 iterations.

Evaluation metrics included absolute and relative error, and sampling time. Clustering quality was assessed using completeness, homogeneity, v-measure, and purity, compared with Fuzzy C-means clustering. Results were averaged over 1,000 random combinations of symptoms.

The dataset from Kaggle ("Disease Prediction Using Machine Learning" by Kaushil268) includes two CSV files for training and testing, each with 133 columns (132 symptoms and 1 prognosis), mapping symptoms to 42 diseases.

We visualized symptom clusters and their relationships to diseases using NetworkX and Pyvis, with graph layout optimized by spring_layout.

## RESULTS

### Evaluation metrics

We aim to assess the accuracy of the calculated probability values obtained from the naive Bayes method by employing two evaluation metrics: absolute and relative errors.

Absolute approximation error is a measure that quantifies the discrepancy between the approximate probability values and the ones obtained using the naive Bayes method. It is calculated as the absolute value of the difference between these two probabilities. For our purposes, let us denote the naive Bayes-calculated probability as $P(X|E)$ and the approximate probability obtained through sampling algorithms as $\hat{P}(X|E)$. In an ideal scenario, we expect the two calculated probability values to be closely approximated, which can be expressed by the Eq. (28) (*Dagum & Luby, 1997*):

$$\hat{P}(X|E) \approx P(X|E). \tag{28}$$

### *Absolute error*

To determine the accuracy of the naive Bayes-calculated probability, we consider the absolute error, which ensures that the approximate probability falls within a certain

**Table 3 Evaluation metrics for different sampling algorithms.**

| Sample no. | Sample size | Approx prob (RS) | Abs error (RS) | Rel error (RS) | Time (RS) | Approx prob (LW) | Abs error (LW) | Rel error (LW) | Time (LW) | Approx prob (GS) | Abs error (GS) | Rel error (GS) | Time (GS) |
|---|---|---|---|---|---|---|---|---|---|---|---|---|---|
| 0 | 316 | 0.532 | 0.032 | 0.060 | 10.142 | 0.510 | 0.010 | 0.020 | 5.348 | 1.000 | 0.500 | 0.500 | 66.977 |
| 1 | 599 | 0.494 | 0.006 | 0.012 | 10.136 | 0.500 | 0.000 | 0.000 | 5.189 | 1.000 | 0.500 | 0.500 | 67.288 |
| 2 | 1,136 | 0.495 | 0.005 | 0.011 | 10.249 | 0.460 | 0.040 | 0.087 | 5.038 | 1.000 | 0.500 | 0.500 | 66.912 |
| 3 | 2,154 | 0.501 | 0.001 | 0.002 | 10.375 | 0.490 | 0.010 | 0.020 | 5.279 | 1.000 | 0.500 | 0.500 | 73.071 |
| 4 | 4,084 | 0.503 | 0.003 | 0.006 | 10.366 | 0.490 | 0.010 | 0.020 | 5.304 | 1.000 | 0.500 | 0.500 | 72.065 |
| 5 | 7,742 | 0.500 | 0.000 | 0.001 | 10.538 | 0.495 | 0.005 | 0.011 | 5.546 | 1.000 | 0.500 | 0.500 | 76.089 |
| 6 | 14,677 | 0.499 | 0.001 | 0.002 | 10.263 | 0.505 | 0.005 | 0.010 | 5.364 | 1.000 | 0.500 | 0.500 | 79.412 |
| 7 | 27,825 | 0.500 | 0.000 | 0.001 | 10.612 | 0.495 | 0.005 | 0.010 | 5.489 | 1.000 | 0.500 | 0.500 | 85.376 |
| 8 | 52,749 | 0.499 | 0.001 | 0.002 | 10.962 | 0.500 | 0.000 | 0.000 | 5.578 | 1.000 | 0.500 | 0.500 | 91.602 |
| 9 | 100,000 | 0.497 | 0.003 | 0.006 | 11.151 | 0.495 | 0.005 | 0.010 | 5.663 | 1.000 | 0.500 | 0.500 | 98.376 |

tolerance level $\epsilon$. Specifically, if the absolute error $|P(X|E) - \hat{P}(X|E)|$ is less than or equal to $\epsilon$, we can conclude that $\hat{P}(X|E)$ lies within the interval $[P(X|E) - \epsilon, P(X|E) + \epsilon]$:

For $P(X|E), \hat{P}(X|E) \in [0,1]$: $\hat{P}(X|E)$ is an approximation for $P(X|E)$ with an absolute error $\leq \epsilon$, if

$$|P(X|E) - \hat{P}(X|E)| \leq \epsilon \quad \text{i.e.,} \quad \hat{P}(X|E) \in [P(X|E) - \epsilon, P(X|E) + \epsilon]. \tag{29}$$

### Relative error

Moving on to the relative error, it measures the difference between the approximate probability and the naive Bayes-calculated probability, divided by the naive Bayes-calculated probability (*Gogate & Dechter, 2012*). By taking the absolute value of this difference, we can assess the relative error. Similar to the absolute error, we aim to ensure that the relative error is within a given threshold $\epsilon$. Consequently, if $\left|1 - \frac{\hat{P}(X|E)}{P(X|E)}\right|$ is less than or equal to $\epsilon$, we can infer that $\hat{P}(X|E)$ falls within the range $[P(X|E)(1 - \epsilon), P(X|E)(1 + \epsilon)]$:

$\hat{P}(X|E)$ is an approximation for $P(X|E)$ with a relative error $\leq \epsilon$, if

$$\left|\frac{1 - \hat{P}(X|E)}{P(X|E)}\right| \leq \epsilon \quad \text{i.e.,} \quad \hat{P}(X|E) \in [P(X|E)(1 - \epsilon), P(X|E)(1 + \epsilon)]. \tag{30}$$

By utilizing these evaluation metrics, we can effectively assess the quality and accuracy of the probability values obtained through the Naïve Bayes algorithm in comparison to those derived from the sampling algorithms.

### Results of evaluation metrics

The tabular data in Table 3 contains a comprehensive analysis of three prominent sampling algorithms: Rejection Sampling (RS), Likelihood Weighting (LW), and Gibbs Sampling (GS). These algorithms were utilized to measure the accuracy of probability

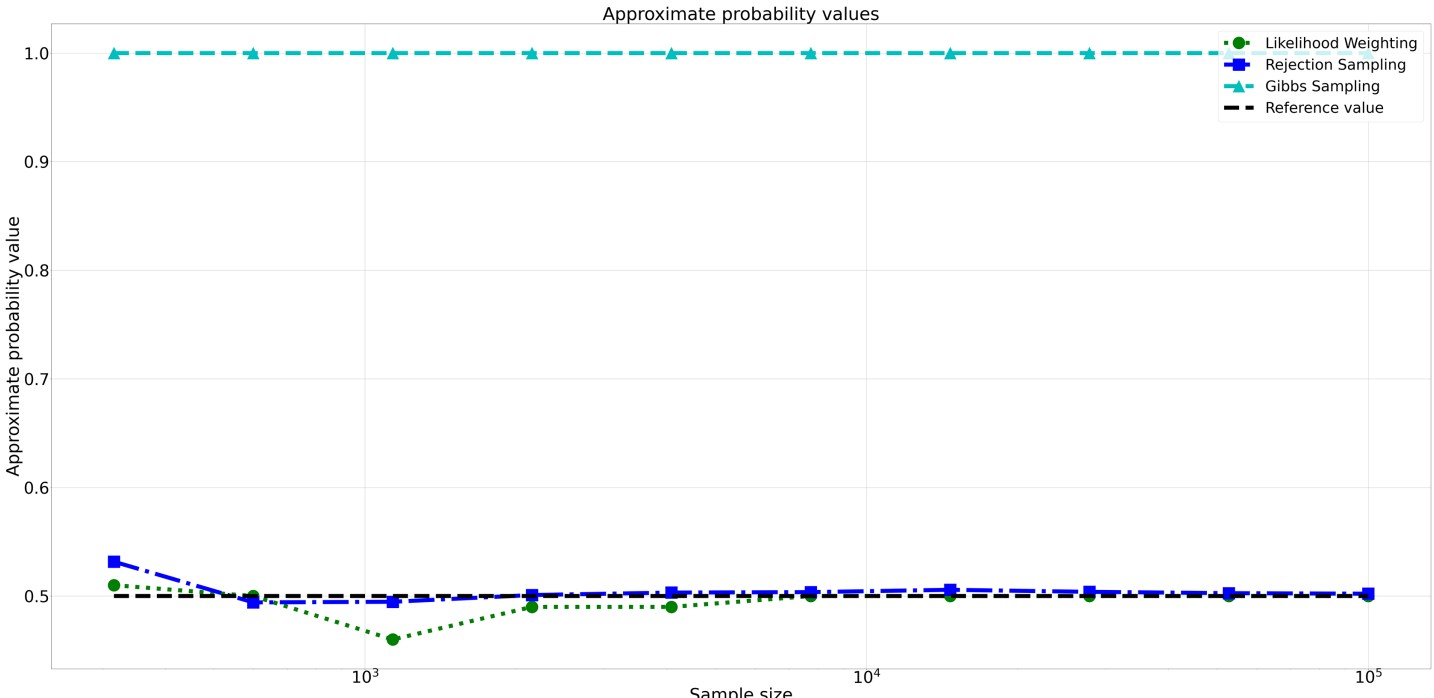

**Figure 5 Approximate probability values for different sampling algorithms.** This figure shows the approximate probability values for Rejection Sampling, Likelihood Weighting, and Gibbs Sampling compared to the reference value across varying sample sizes. Descriptions: Rejection Sampling (RS): Represented by blue squares with a dashed line. Likelihood Weighting (LW): Represented by green circles with a dotted line. Gibbs Sampling (GS): Represented by cyan triangles with a dashed line. Reference value: Represented by a black dashed line.

values calculated using the naive Bayes theorem. The evaluation involved comparing the probability values obtained from the Naïve Bayes theorem with those generated by the sampling algorithms, namely rejection sampling, likelihood weighting, and Gibbs sampling. This comparison allowed us to assess how well the naive Bayes algorithm estimates the true probabilities. The analysis was conducted using 10 different samples of varying sizes, providing valuable insights into the effectiveness and reliability of the Naïve Bayes theorem in estimating probabilities for the given dataset.

The pattern of calculated approximate probability values of 10 samples is shown in Fig. 5. Delving into the assessment of absolute errors (Abs Error), the probability values calculated using the Naïve Bayes algorithm exhibited varying degrees of absolute error when compared to sampling methods, with both Rejection Sampling and Likelihood Weighting demonstrating significantly lower errors compared to Gibbs Sampling. Rejection Sampling consistently achieved absolute errors ranging from 0.001 to 0.032, showcasing its ability to closely approximate the true probabilities. Similarly, Likelihood Weighting displayed impressive accuracy, with absolute errors ranging from 0.000 to 0.040. In contrast, Gibbs Sampling consistently produced remarkably higher absolute errors, all fixed at 0.500, indicating limitations in its ability to accurately approximate the true probabilities. You can observe the absolute errors generated by the sampling algorithms in Fig. 6.

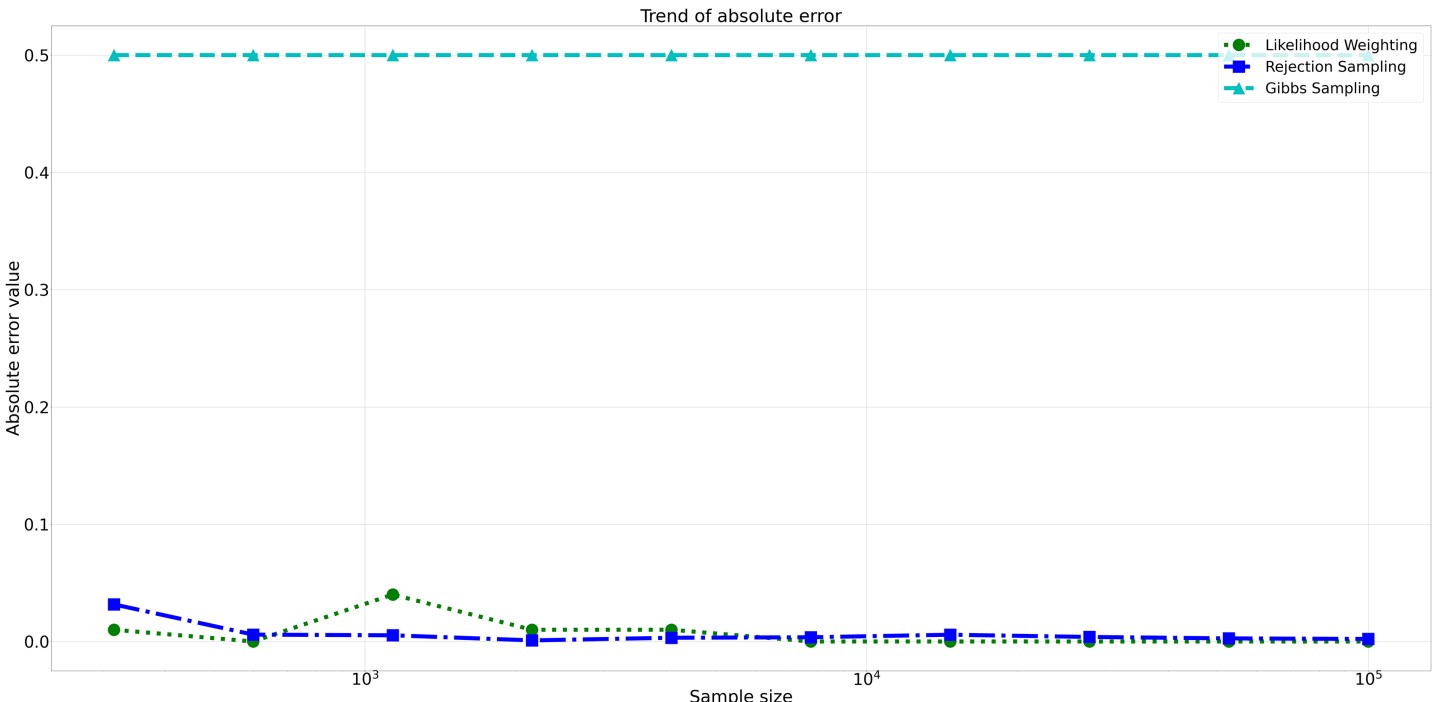

**Figure 6** **Absolute errors for different sampling algorithms.** This figure illustrates the absolute error values for Rejection Sampling, Likelihood Weighting, and Gibbs Sampling across varying sample sizes. Descriptions: Rejection Sampling (RS): Represented by blue squares with a dashed line. Likelihood Weighting (LW): Represented by green circles with a dotted line. Gibbs Sampling (GS): Represented by cyan triangles with a dashed line.

Furthermore, analyzing the relative errors (Rel Error) in Fig. 7, we observe similar trends to the absolute errors. Rejection Sampling and Likelihood Weighting consistently exhibited notably lower relative errors, spanning from 0.001 to 0.060 and 0.000 to 0.087, respectively. On the other hand, Gibbs Sampling persisted with higher relative errors fixed at 0.500 across all sample sizes, indicating its limitations in accurately approximating the true probabilities.

Considering computational efficiency, both Rejection Sampling and Likelihood Weighting outperformed Gibbs Sampling in terms of time taken. Rejection Sampling consistently displayed commendable efficiency, with an average execution time of approximately 10 s. Likelihood Weighting demonstrated similar efficiency, requiring an average execution time of approximately 5 s for all sample sizes. Conversely, Gibbs Sampling exhibited significantly longer execution times, starting at approximately 67 s for the smallest sample size and increasing to nearly 98 s for the largest sample size as shown in Fig. 8.

## Assessment of clustering quality
### Completeness

Completeness checks if all data points of a class are grouped into a single cluster (*Lu & Uddin, 2024*). It is considered complete when each class is entirely within one cluster. The formula is:

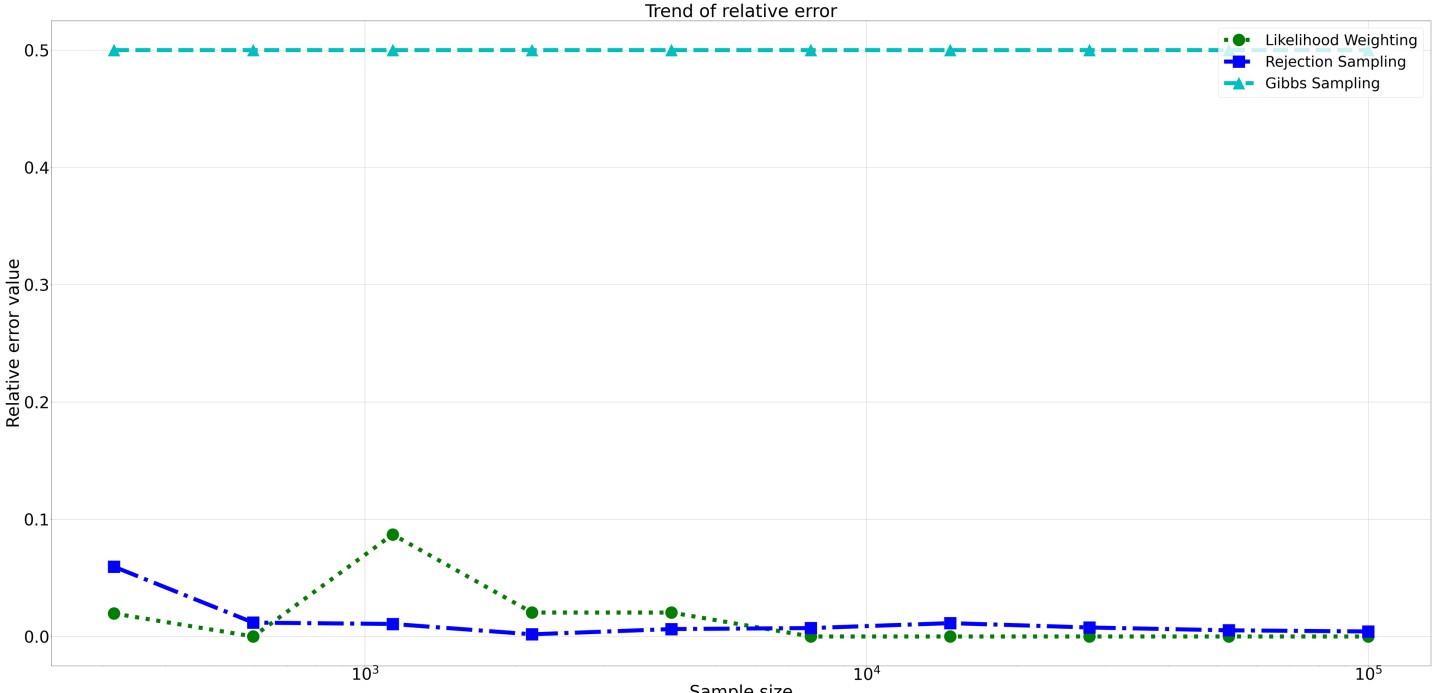

**Figure 7 Relative errors for different sampling algorithms.** This figure displays the relative error values for Rejection Sampling, Likelihood Weighting, and Gibbs Sampling across varying sample sizes. Descriptions: Rejection Sampling (RS): Represented by blue squares with a dashed line. Likelihood Weighting (LW): Represented by green circles with a dotted line. Gibbs Sampling (GS): Represented by cyan triangles with a dashed line.

$$\text{Completeness} = 1 - \frac{H(C|K)}{H(C)} \tag{31}$$

where $H(C|K)$ is the conditional entropy of the class distribution given the cluster assignments, and $H(C)$ is the entropy of the class distribution.

Completeness measures how well clustering algorithms group all symptoms of each disease into single clusters. A higher score indicates successful grouping of symptoms of the same disease together, preserving the disease's symptom profile.

### *Homogeneity*

Homogeneity compares clustering outcomes to a ground truth, considering a cluster homogeneous if it contains data points from a single class (*Lu & Uddin, 2024*). The formula is:

$$\text{Homogeneity} = 1 - \frac{H(K|C)}{H(K)} \tag{32}$$

where $H(K|C)$ is the conditional entropy of cluster assignments given disease labels, and $H(K)$ is the entropy of cluster assignments.

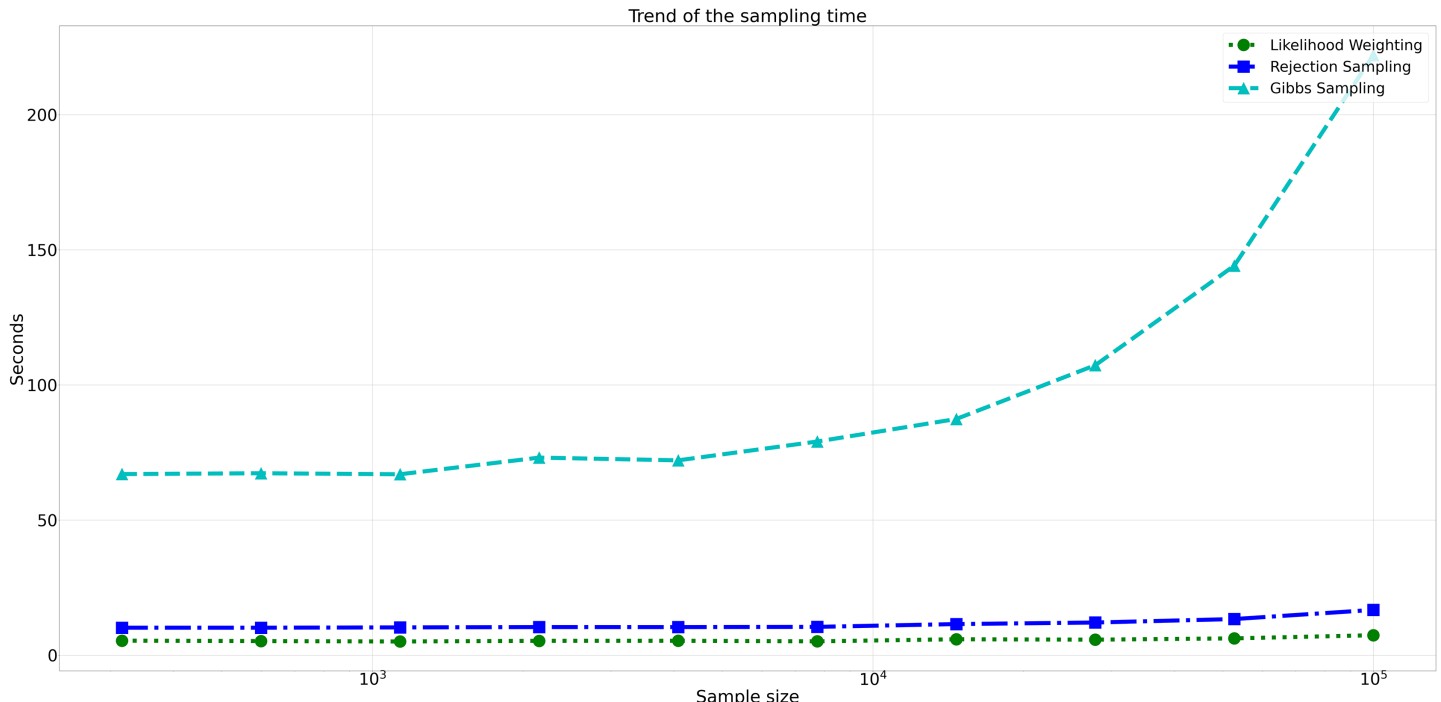

**Figure 8 Trend of sampling time for different sampling algorithms.** This figure shows the computational efficiency of Rejection Sampling, Likelihood Weighting, and Gibbs Sampling across varying sample sizes. Descriptions: Rejection Sampling (RS): Represented by blue squares with a dashed line. Likelihood Weighting (LW): Represented by green circles with a dotted line. Gibbs Sampling (GS): Represented by cyan triangles with a dashed line.

Homogeneity measures if clusters contain symptoms predominantly from one disease. High homogeneity indicates clusters mostly composed of symptoms from a single disease, though this may be challenging due to overlapping symptoms across diseases.

### V-measure

V-measure is the harmonic mean of homogeneity and completeness (*Lu & Uddin, 2024*). V-measure provides a balanced evaluation by considering both the purity of the clusters and the extent to which all symptoms of a disease are grouped together. This score helps us compare the overall performance of the Proposed Algorithm and Fuzzy C-means clustering algorithms in capturing the underlying disease structure in the data.

$$V - \text{measure} = 2 \times \frac{\text{Homogeneity} \times \text{Completeness}}{\text{Homogeneity} + \text{Completeness}}. \tag{33}$$

### Purity

Purity measures how well clusters contain data points from a single class. High purity indicates that clusters are mostly composed of data points from the same class (*Dong et al., 2024*). The formula is:

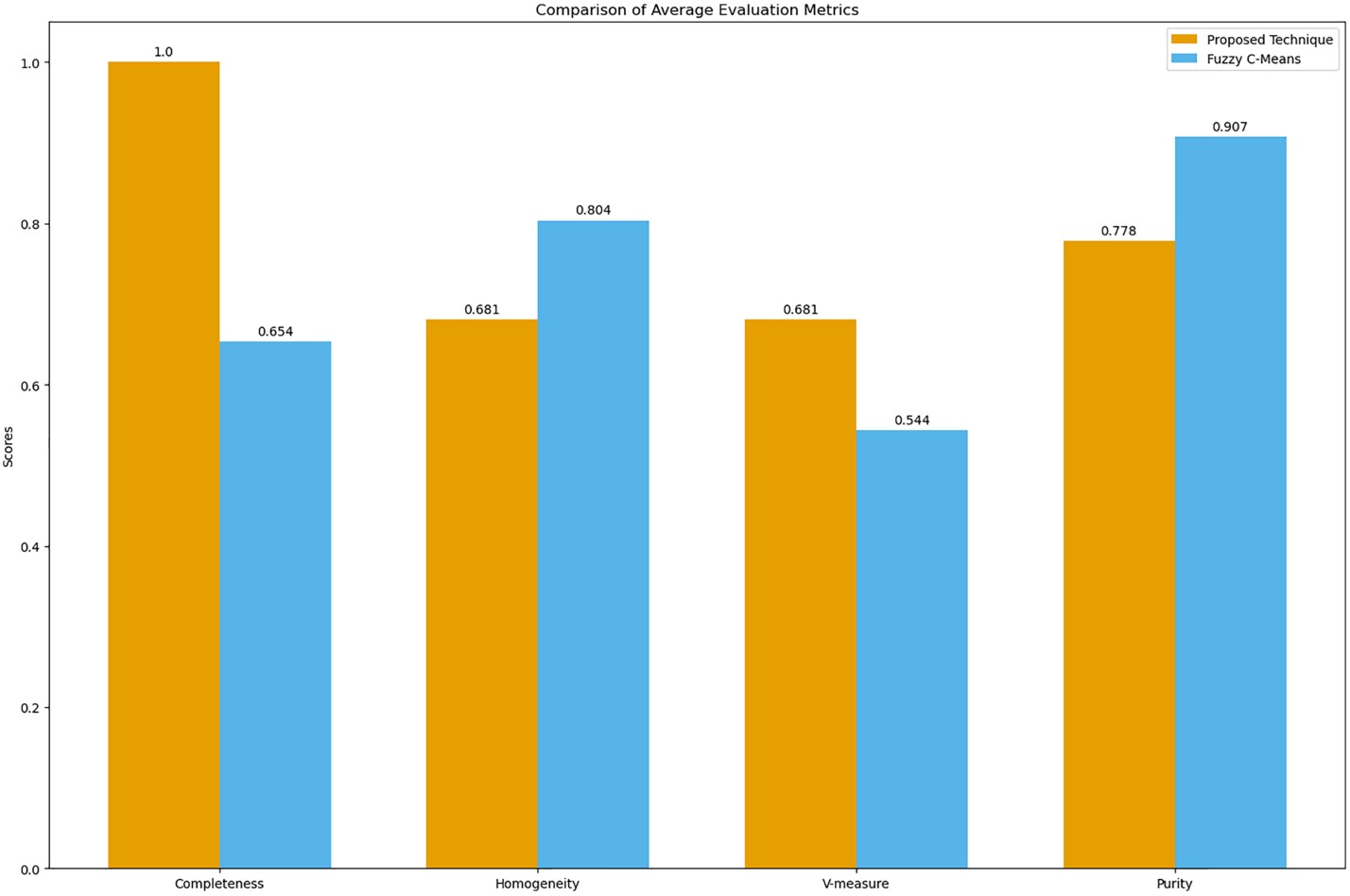

**Figure 9 Comparison of average clustering evaluation metrics.** This figure illustrates the average scores for completeness, homogeneity, V-measure, and purity for the proposed technique and Fuzzy C-means.

$$\text{Purity} = \frac{1}{N} \sum_k \max_j |c_k \cap t_j| \qquad (34)$$

where $N$ is the total number of data points, $c_k$ is the set of data points in cluster $k$, and $t_j$ is the set of data points in class $j$.

Purity assesses how well each cluster contains data points primarily from one class. It assigns each cluster to its most frequent class and counts the correctly assigned data points. A high purity score means most data points in each cluster belong to the same class. However, overlapping symptoms across multiple diseases might lower purity scores.

### Comparative analysis

To validate our proposed algorithm, we computed the completeness, homogeneity, V-measure, and purity for both the proposed algorithm and fuzzy C-means. This comparison allows us to benchmark the performance of our algorithm against the state-of-the-art Fuzzy C-means algorithm.

**Table 4 Comparison of average clustering evaluation metrics for proposed technique and fuzzy C-Means algorithms.**

| Metric | Proposed technique | Fuzzy C-Means |
|---|---|---|
| Completeness | 1.0 | 0.654 |
| Homogeneity | 0.681 | 0.804 |
| V-measure | 0.681 | 0.544 |
| Purity | 0.778 | 0.907 |

The bar graph visually compares the average evaluation metrics for the proposed technique and fuzzy C-means (see Fig. 9).

The results in Table 4 indicate that the proposed technique achieves perfect completeness (1.0), which suggests that it effectively groups all symptoms of the same disease together. The homogeneity (0.681) and purity (0.778) scores for the proposed technique are lower compared to Fuzzy C-means, which scores 0.804 for homogeneity and 0.907 for purity. This is expected because our main goal was to propose a technique that generates clusters of overlapping symptoms. Therefore, the homogeneity and purity are lower. The higher V-measure for the proposed technique (0.681) compared to Fuzzy C-means (0.544) indicates that our algorithm achieves a better balance between completeness and homogeneity, effectively capturing the underlying structure of the data despite the complexity of overlapping symptoms.

## DISCUSSION

This study analyzed the use of the naive Bayes algorithm for clustering uncertain overlapping symptoms of multiple diseases in clinical diagnosis. Utilizing a conditional probability matrix and a Bayesian network, the algorithm effectively estimated disease cluster probabilities based on observed symptoms. Comparison with likelihood weighting, rejection, and Gibbs sampling highlighted the superior accuracy and computational efficiency of the naive Bayes algorithm. Gibbs sampling produced larger errors due to its calculation based on neighboring nodes, unlike the naive Bayes theorem, which considers only the disease and symptom nodes.

The assessment of clustering quality provided valuable insights. The completeness score of 1.0 demonstrated the algorithm's effectiveness in grouping symptoms of the same disease, capturing the underlying disease structure. While homogeneity and purity scores were lower than those of the Fuzzy C-means algorithm, they remained within acceptable ranges, reflecting the complexity and overlap of symptoms. The higher V-measure score indicated a better balance between completeness and homogeneity.

The evaluation metrics—completeness, homogeneity, V-measure, and purity—offered a balanced view of performance. Although Fuzzy C-means showed higher homogeneity and purity, it struggled with completeness and V-measure, critical for understanding overlapping symptoms. The proposed algorithm's superior V-measure suggests its robustness in managing symptom overlap and uncertainty, making it more suitable for clinical diagnosis applications.

# CONCLUSION

This study highlights the promising application of the naive Bayes algorithm for clustering uncertain overlapping symptoms in clinical diagnosis. Using a conditional probability matrix and Bayesian network, the algorithm effectively estimated disease cluster probabilities, offering valuable insights for clinical decision-making. Compared to benchmark sampling algorithms, the naive Bayes approach demonstrated superior accuracy and computational efficiency, suggesting its potential to enhance clinical diagnosis processes and patient outcomes.

The high completeness and V-measure scores validated the algorithm's effectiveness in managing symptom overlap and uncertainty. Despite lower homogeneity and purity scores compared to Fuzzy C-means, the proposed algorithm's robust performance makes it suitable for real-world clinical applications.

Future research could refine the algorithm and incorporate additional data sources to expand its applicability across diverse clinical contexts. Enhancements might include more sophisticated probabilistic models and alternative similarity measures for categorical data. The algorithm's adaptability also holds potential for applications in other fields, such as clustering movies and shows based on hidden associations for personalized recommendations.

## Funding
The authors received no funding for this work.

## Competing Interests
The authors declare that they have no competing interests.

## Author Contributions
- Asif Ali Wagan conceived and designed the experiments, performed the experiments, analyzed the data, performed the computation work, prepared figures and/or tables, authored or reviewed drafts of the article, designed the database to store the symptom information, and approved the final draft.
- Shahnawaz Talpur analyzed the data, authored or reviewed drafts of the article, and approved the final draft.
- Sanam Narejo performed the experiments, authored or reviewed drafts of the article, and approved the final draft.

## Data Availability
The dataset is available at Kaggle: https://www.kaggle.com/datasets/kaushil268/disease-prediction-using-machine-learning. The raw data contains comprehensive details for all instances used in our analysis, including the symptoms and prognoses. This dataset, created by Kaushil Mangaroliya, comprises 132 symptoms and can predict 42 different

types of diseases. The dataset is split into two files: one for training and one for testing machine learning models.

The code is available in the Supplemental Files.

## Supplemental Information

Supplemental information for this article can be found online at http://dx.doi.org/10.7717/peerj-cs.2315#supplemental-information.

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
