# Peer review of "Clustering uncertain overlapping symptoms of multiple diseases in clinical diagnosis"

_PeerJ Computer Science, doi:10.7717/peerj-cs.2315_

## Round 0.1 · original submission · Major Revisions

Based on the external reviews, a major revision is needed for further consideration. The authors should revise the manuscript carefully.

Reviewer 1 ·

Basic reporting

In the manuscript titled 'Clustering Uncertain Overlapping Symptoms of Multiple Diseases in Clinical Diagnosis,' Asif Ali et al. described a new clustering algorithm that considers the precise weighting of each symptom in every disease. Overall, the work is promising and well presented. Some specific comments are as follows:
1. The manuscript may need slight English proofreading to fix sentences like 'In various fields, including medical science, datasets characterized by uncertainty are generated.'
2. I would recommend adding figure legends to describe the figures, especially Figures 3 and 4, to make them more intuitive.
3. I would recommend including titles and brief descriptions for the supplementary tables and referencing them in the main text.

Experimental design

The title of the manuscript suggests the clustering of multiple diseases. However, only subtypes of liver infections are tested. I would recommend testing your algorithm on diseases other than liver infections to demonstrate its broad applicability.

Validity of the findings

no comment

Additional comments

Asif Ali et al. mentioned in the manuscript that 'the disease symptoms dataset is available at Kaggle: kaggle.com/datasets/kaushil268/disease-prediction-using-machine-learning.' However, I cannot access the dataset at the URL provided.

Reviewer 2 ·

Basic reporting

Some sections of the manuscript could benefit from better organization and clarity of presentation. For instance, the 4.x.x section combines multiple analyses, making it challenging to follow the narrative. Making it clear and rearranging the content could enhance readability and comprehension.

While a literature review is provided, a more critical analysis of existing overlapping clustering techniques for uncertain data is needed. Clearly positioning the proposed work concerning previous approaches and highlighting its novelty and advantages would help establish its significance and contribution to the field.

Experimental design

Details regarding the experimental setup, such as dataset characteristics (size, number of diseases, symptom distributions), parameter settings, and implementation details, are missing. Providing this information is crucial for reproducibility and assessing the generalizability of the results.

Validity of the findings

The evaluation section primarily focuses on assessing the accuracy of the Naive Bayes probability calculations against sampling algorithms. However, there is a lack of comprehensive evaluation of the clustering performance itself. Including quantitative metrics (e.g., cluster purity, completeness, homogeneity) and comparing the proposed approach against established baselines or state-of-the-art methods would significantly strengthen the evaluation.

---

## Round 0.2 · accepted · Accept

Based on the external reviews, the manuscript is now ready for acceptance.

Reviewer 1 ·

Basic reporting

All my concerns have been addressed.

Experimental design

All my concerns have been addressed.

Validity of the findings

All my concerns have been addressed.

Reviewer 2 ·

Basic reporting

Authors addressed all the suggested changes.

Experimental design

Authors addressed all the proposed changes.

Validity of the findings

Authors addressed all the proposed changes.